# The Role of the Avatar in Gaming for Trans and Gender Diverse Young People

**DOI:** 10.3390/ijerph17228617

**Published:** 2020-11-20

**Authors:** Helen Morgan, Amanda O’Donovan, Renita Almeida, Ashleigh Lin, Yael Perry

**Affiliations:** 1College of Science, Health, Engineering and Education (SHEE), Murdoch University, Perth, WA 6150, Australia; a.odonovan@murdoch.edu.au (A.O.); renita.almeida@murdoch.edu.au (R.A.); 2Telethon Kids Institute, The University of Western Australia, Perth, WA 6009, Australia; ashleigh.lin@telethonkids.org.au (A.L.); yael.perry@telethonkids.org.au (Y.P.)

**Keywords:** avatar, gaming, trans and gender diverse, gender identity, gender questioning, serious games

## Abstract

A significant proportion of trans and gender diverse (TGD) young people report membership of the gaming community and resultant benefits to wellbeing. To date their experiences and needs regarding a key feature of games, the avatar, are largely unexplored, despite increasing interest in the therapeutic role of avatars in the general population. The aim of this study was to better understand the role of the avatar in gaming, its impact on TGD young people’s mental health, and their unique needs regarding avatar design. N = 17 TGD young people aged 11–22 years (M = 16.3 years) participated in four focus groups. A general inductive approach was used to thematically analyze the transcribed data. TGD young people report considerable therapeutic benefits of using avatars with positive mental health implications. Importantly, TGD young people use avatars to explore, develop and rehearse their experienced gender identities, often as a precursor to coming out in the offline world. They also report negative experiences of feeling excluded due to the constraints of conventional notions of gender that are widely reflected in game design. Participants described simple design features to better reflect gender diversity, such as increased customization. Such changes would facilitate the positive gains reported by participants and better reflect the diversity of young people who use games. The findings have important implications for both recreational and serious or therapeutic game design.

## 1. Introduction

Individuals whose gender identity is incongruent with the sex assigned to them at birth identify as transgender or gender diverse (TGD). The prevalence of gender diversity has increased steadily in recent decades, sitting at estimates of 1.2–2.7% for TGD adolescents in the general population [1,2,3]. Adolescence is customarily a time of significant developmental change and a critical time for mental health; half of all mental health disorders emerge during this period [4] and approximately three-quarters of all mental disorders are present by the mid-twenties [4,5]. The intersection of gender non-conformity with adolescence in a heteronormative world can exacerbate the common challenges of adolescence. TGD youth consequently report a raft of additional challenges, including increased stigmatization, familial and peer rejection, mental health difficulties and barriers to accessing support services [1,6,7,8,9]. Consequently TGD young people are at particularly high risk of developing mental health difficulties. Nearly three quarters of Australian TGD young people report having a depression or anxiety diagnosis, nearly eighty percent report self-harming, and nearly half have attempted suicide [10].

In the face of these isolating contexts, one way in which TGD young people report positive mental health gain is through use of the internet, including gaming [10]. Over 57% of Australian TGD young people reported using video games, and over 54% indicated they did so to make themselves feel better when experiencing low or negative mood [10]. Gaming has been shown to be a positive way of coping with distress and to increase psychological wellbeing in cisgender (who identify with their birth-assigned sex) populations [11,12,13]. Although scant, TGD studies reporting on online behaviors indicate that gaming can be experienced as a positive psychological tool to facilitate self-awareness and expression of their experienced gender in a relatively safe environment [14,15]. Many of these games use avatars, a virtual world digital self-representation which is often customizable. Furthermore, anecdotal reports of TGD users’ ability to adapt avatars to represent a desired or real identity, aligned with their experienced gender identity, confers mental health benefits such as increases in positive mood and self-esteem [14,16,17,18].

Within the broader gaming research, the avatar is noted repeatedly as one of the most positive aspects of online physical and mental health interventions [19,20,21]. Reasons provided include the ability to represent the self or aspects of the self and expressing emotions that would be difficult to communicate verbally in person [20]. Exploration of the use of avatars in digital mental health interventions is a burgeoning area, reflecting improvements in individual functioning across a range of areas including improved social functioning in young adults with autism [22], reduced depressive symptoms in adolescents [23] and reduced frequency and intensity of auditory hallucinations in people with schizophrenia [24]. Importantly, inhabiting an avatar that represents the idealized or experienced self appears to be therapeutic in addition to the intended benefits of the intervention itself [20].

Finally, serious gaming (i.e., the use of games for purposes such as learning or health improvements) research involving sexually and gender diverse youth indicates that the utility of an intervention is influenced by the capacity of the avatar for user representation [25]. Given these findings, and the lack of dedicated research to date, a better understanding of how TGD young people experience the avatar in gaming has important implications for both recreational and serious game design. In turn, more suitable game construction may facilitate better mental health outcomes for TGD young people. Accordingly, the aim of this study is to better understand the role of the avatar in gaming for TGD youth, its impact on TGD youths’ mental health, and their unique needs regarding avatar design.

## 2. Materials and Methods

### 2.1. Recruitment

TGD young people who were aged 11–22 years and able to attend focus groups in the Perth metropolitan area were eligible for the study. Parental written informed consent was required for participants under 14 years of age but waived for older participants who met capacity requirements for self-consent. Convenience sample recruitment occurred via study promotion in local and university LGBT+ (lesbian, gay, bisexual and trans) services, through social media and an existing longitudinal gender diversity cohort study. All young people provided assent/consent and participated fully in the groups with no withdrawals from the study. The studies were approved by University of Western Australia (RA.4.20.4242) and Murdoch University (2018/116, 2018/200) human research ethics committees.

### 2.2. Data Collection

This study was adjunct research to a broader project focused on adapting an online serious game to prevent depression in TGD young people. Participants took part in two closed focus groups totaling four hours’ duration designed to explore attitudes towards online experiences and digital health interventions and to provide feedback on the serious game being adapted. The data were transcribed, themes examined for saturation, and additional questions developed relating to avatar use for further exploration in another two-hour group, independent of the original research. Five of the original participants returned to contribute further. All questions were developed by researchers with collective expertise in digital health, TGD youth mental health and qualitative research.

Upon group commencement, participants completed a demographic survey and facilitators established group rules regarding confidentiality, respect and safety. Support service information and access pathways were provided. Attendees were remunerated at each focus group. Two members of the research team (comprising DWT, HM, PS, or YP) facilitated each group. Three identify as cisgender females, and one identifies as TGD, all have research or clinical experience in working with TGD young people. Two researchers were known to some participants, however, there were no other established facilitator–participant relationships Groups were held at a community LGBTIQ+ youth drop-in space and at the Murdoch Psychology Clinic with nobody else present. All focus groups were audio recorded and field notes taken during and immediately afterwards to ensure rigor.

### 2.3. Analysis

The aim of the study was to explore key themes in the focus group responses without the imposition of pre-determined premises and develop a model reflecting the structure of participants’ experiences based on the data. A general inductive approach (GIA), similar to grounded theory and phenomenological approaches, was used to analyze the data and build understanding from observation rather than testing a priori hypotheses [26]. The focus groups were audio-recorded and transcribed verbatim, and the transcripts were then checked against the audio recordings prior to analysis. The transcripts were de-identified by removing participants’ names and any other potentially identifying details. Key themes were identified and coded using NVivo qualitative data analysis software (Version 11, QSR International Pty Ltd., Melbourne, Australia). After repeated listening of the audio recordings and reading of the transcripts to consider in-text meaning without prior expectation, the following steps, as suggested by Thomas [26], were taken by HM: (1) specific segments of information were identified; (2) segments were labelled to create categories; (3) similar categories were clustered to reduce overlap and redundancy; (4) a model incorporating the most important categories was created in consultation with co-author AOD.

Quality procedures were used to enhance the trustworthiness of the study findings [27], such as member checks at the end of each focus group, copies of the full transcript made available to participants for comment and a summary of key themes provided to all participants prior to the final round of data analysis for approval and further comment. Other techniques such as a preliminary literature review, audio recording of the groups, and field notes were employed to ensure rigor. The consolidated criteria for reporting qualitative research (COREQ) checklist (see Appendix A) was used to guide comprehensive reporting of findings and is included as a supplemental file [28]. COREQ is a 32-item checklist used for comprehensive and explicit reporting of qualitative studies that utilise in-depth interviews and focus groups [28].

## 3. Results

Seventeen TGD young people participated ranging from 11–22 years (*M* = 16.3 years, *SD* = 2.76). Participants self-identified as “male” or “trans male” (*n* = 11), “female” (*n* = 1), agender or non-binary (*n* = 3) and bi-gender (male and agender; male and non-binary *n* = 2). Two young people reported they had been assigned male at birth and fifteen assigned female at birth.

Two overarching domains emerged from the analysis, specifically: (Section 3.1) Avatar Use and Function; and (Section 3.2) Experiences and Perspectives of Participants Regarding Avatar Design. The primary themes and subthemes for both domains are reported in order of prevalence and described below. Accompanying numbered illustrative quotes are outlined in Table 1 and Table 2.

### 3.1. Avatar Use and Function

The majority of uses and functions expressed by participants related to the conceptualization and actualization of users’ experienced gender through the embodiment of the avatar. Specifically, the theme of creating an avatar which closely reflected the user’s gender identity, and the theme that this process facilitated gender identity consolidation for the participant, particularly in the early stages of gender questioning and experimentation, were equally prevalent.

#### Avatar as a Reflection of Experienced Gender

Creation of avatar as experienced gender.

For many participants, the creation or selection of an avatar that reflected their experienced gender was a first, or formative, step in privately acknowledging their gender identity before they came out in the real (offline) world. The virtual embodiment of their gender identity was deemed to be a safe and anonymous means of testing out experienced gender in preparation for more public representations of the self (#1).

Creating avatar facilitates gender identity consolidation.

Once young people were using avatars that reflected their experienced gender, they also described an ongoing process of experimenting with, refining, and finally consolidating this identity in the relative safety of the gaming world (#2). Thus, the theme of the avatar played a crucial role in refining young people’s sense of gender identity before introducing elements of identity in real world interactions. Specific games were cited as being particularly useful for facilitating this process by providing diverse possibilities (e.g., clothing and body shape options) to curate a unique avatar reflective of the user’s gender expression. Some games also facilitated determination of other aspects such as preferred name and use of pronoun/s.

Positive emotional impact of using avatar in experienced gender.

Participants described positive emotions when playing as an avatar created in their experienced gender. When reflecting on these experiences, the pairing of enacted experienced gender with positive emotion affirmed their gender identity, particularly in the fledgling stages of social transition. For some, avatars were the only outlet for their gender expression before coming out, indicating the potentially therapeutic role avatar curation played during an emotionally challenging period. Positive emotions were also experienced when avatars were acknowledged and accepted in their experienced gender in-game, for example, through game dialogue or by other players online (#3). Underpinning this subtheme was the concept of acceptance, both of the self and by others.

Participants also noted that creating an avatar in their experienced gender was often a therapeutic task in its own right, undertaken to offset negative emotions, and could occur without intent to play the game further. Participants also used a combination of their current and aspired appearance during curation as a deliberately employed strategy, particularly in the earlier stages of their gender identity journey. This approach minimized negative emotions about the ongoing process towards complete realization of experienced gender and maintained positivity for the future.

### 3.2. Experiences and Perspectives of Participants Regarding Avatar Design

Participants spoke about their experiences of designing or choosing avatars and the features they would like to see incorporated into game design.

#### 3.2.1. Customization

Customization is key and facilitates gender identity expression.

The importance of being able to express one’s gender identity through avatar customization was the strongest focus group theme of all and cited as the most important design feature. This included body shape, hair, facial features, accessories and the ability to select and mix both conventionally binary gendered features within one avatar. Reflecting and consolidating one’s gender identity through customization was described as a positive therapeutic process and some described this process as key to their own realization that they could choose to reflect their experienced gender identity in the offline world as well. This suggests that while avatar customization can be a diversionary activity, it also facilitates mental processing regarding the journey of gender identity expression (#4 and 5).

Customization determines playability/purchase.

The ability to customize the avatar was pivotal when choosing to purchase a game. Several key games, purchased specifically for their customization features, were cited (#6).

#### 3.2.2. Avatar Gender Identity

Customizable or diverse options for pronouns

Participants across the focus groups voiced strong support for a more diverse range of customizable pronouns to better reflect their gender identity (#7). Highlighted within this subtheme was the feeling of inclusion, and the positive emotional impact upon users that more thoughtful design would generate. As so many games require players to inhabit a binary gender from the outset, even the seemingly small detail of customizable pronouns was deemed meaningfully inclusive.

Non-binary options

Participants voiced a specific need for identities that fall outside of the binary, stressing the positive impact a more thoughtful approach to inclusivity would have upon users (#8). They also reflected upon how alienating it was to feel a disconnect with an avatar when restricted by conventional notions of gender.

#### 3.2.3. Negative Experiences: Exclusion and Lack of Connection

Lastly, participants spoke about negative experiences of using an avatar, specifically, feeling excluded from playing games with non-customizable or exclusively binary avatars only and experiencing a lack of connection with the avatar (#9).The youngest participants also highlighted difficulties with imposition of birth sex-assigned avatars on school and educational websites and apps and the negative impact this had on them (#10).

## 4. Discussion

In this study we sought to better understand how TGD young people experience the avatar in gaming using a convenience sample, given the dearth of research in this field to date. Findings illustrate the ways in which TGD young people use avatars, as well as their desires and requirements for avatar design. Importantly, this is the first dedicated study to find that TGD young people create or select an avatar to reflect their experienced gender and that this use of the avatar facilitates gender identity expression and consolidation with powerful mental health benefits. These findings support anecdotal clinical reports [10,14,15] and the online reports and reflections of some TGD gamers [16,17,18,29] and significantly extend our understanding of their experiences.

For adolescents an online presence can be key to identity formation and expression during an important developmental period [30]. Moreover, given the potential risks to TGD young people of living as one’s experienced gender identity in real life, online media can provide a vital opportunity for users to explore, develop and rehearse their identities in a relatively safe environment. For many, online experiences are a testing ground for gender identity exploration and coming out before expansion into the offline world [31,32]. Our study adds to this body of research in finding that avatar use, specifically, meaningfully facilitates the vital practices of exploration, formation and rehearsal across the stages of gender identity development, often before coming out in the real world.

Theoretically, our participants’ experiences with avatars were congruent with contemporary models of transgender identity development, particularly with regard to the interpersonal and intrapersonal processes of exploring, affirming, and integrating one’s identity [33]. Devor’s model outlines a further interpersonal process, “witnessing”, which aids individuals in progressing through stages of gender identity formation by having one’s sense of self accurately reflected back by others who do not share such similarity [34]. Our participants reported experiencing validation and positive emotions when other players unquestioningly accepted and affirmed their avatar’s gender in-game.

That participants reported powerful positive emotions when inhabiting an avatar that reflected their experienced gender identity, and negative feelings when unable to choose gender, aligns with research conducted on avatar use in the general population. Individuals who use a customized avatar exhibit more physiological arousal, suggesting they experience an affinity with it that augments their experience, and report feeling the avatar more closely reflects their identity [35,36]. As Frow states, the avatar is a crucial site of “affective investment” [37] (p. 361) in gaming for players, and has become an embedded and elaborate tool for self-expression through customization across the gaming landscape [38].

Considering the increased negative life events and barriers to accessing support experienced by TGD young people, and the protective effect of gaming for TGD mental health, the potential of more inclusive avatar design extends past the issue of enhanced playability and into the realms of facilitating a valuable therapeutic process for many TGD young people [10]. This speaks to Rehm et al.’s [20] assertion that the ability to inhabit an avatar and have it represent the idealized or experienced self appears to be therapeutic in addition to the intentional benefits of the intervention in which the avatar is located.

To our knowledge, this is the first dedicated study to determine what TGD young people require regarding serious game avatar design by exploring recreational gaming experiences. The findings of this study can further inform serious game design given TGD young people are likely to be users of digital mental health interventions. Participants outlined a set of proposals regarding more inclusive avatar design with three key recommendations. Firstly, users should not be constricted by binary notions of gender which coerce them into a binary gender choice from the outset of the game. The freedom to combine all possible avatar customization options allows users to fluidly tailor their avatar to mirror their experienced gender identity. Participants were clear this design feature should be embedded throughout the game to facilitate evolution and consolidation of gender identity over time. Secondly, the need for diversity in character representation extended beyond visual features to language. Participants requested diversity in pronoun selection from the outset of the game which also determines subsequent interactions such as in-game dialogue. Such interactions were cited as powerfully validating of their experienced gender identity and consequently held therapeutic value. Thirdly, game design must account for the spectrum of gender diversity, specifically the inclusion of non-binary customization options. This is particularly important given almost half of TGD young people in Strauss et al.’s study described their gender in non-binary terms [10].

Accordingly, consideration of TGD gamers’ needs does not require major revision, but greater recognition of human diversity and responsive expansion of simple key elements of design, primarily, to facilitate a more inclusive experience. This was a small-scale exploratory study as is common in qualitative research, however, the findings are limited by the somewhat homogenous sample which was not representative of all TGD young people who game. Moreover, recruitment was geographically restricted to one city, and most participants were assigned female at birth, further limiting generalizability.

The findings are important to consider regarding the evolving development of serious games which are increasingly pertinent to accessing youth populations. Importantly, there is a current initiative, SPARX-T, to develop an online intervention tailored specifically to the needs of TGD youth in preventing depression (Perry et al., in preparation) [39]. TGD young people assert that game-based digital mental health interventions need to be trans-affirmative and gender inclusive in content to be appealing [40]. The findings of this study can further inform serious game design given TGD young people are likely to be users of e-mental health interventions.

The findings also indicate a number of important avenues to be pursued in future research. In particular, the role of the avatar in gender identity exploration and consolidation holds intriguing therapeutic possibilities and more nuanced understanding of how this process unfolds is needed. Further exploration of how the use of the avatar impacts on offline or real world experiences of realizing experienced gender is also worthy of pursuit, particularly how the experiences of avatar use reported by our participants impacted the potentially challenging process of coming out in the offline world. Similarly, the ability of the avatar to influence mood and elicit powerful emotional responses is noted in the wider field of avatar research, however further research on how these functions can be therapeutically harnessed for TGD populations is warranted.

## 5. Conclusions

TGD young people report considerable therapeutic benefits from avatar use although they also describe feeling excluded and under-represented within mainstream gaming design, primarily due to the application of conventional notions of gender. Importantly, TGD young people use avatars to explore, develop and rehearse their experienced gender identities, often as a precursor to coming out in the offline world. TGD young people stipulated predominantly simple game design features, such as greater avatar customization, to better reflect gender diversity. Such changes would likely facilitate the positive gains reported by participants and better reflect the heterogeneity of young people who game. The findings establish a detailed understanding of how TGD young people experience the avatar in gaming and indicate exciting avenues for future development, particularly regarding the therapeutic promise the avatar holds. The findings have important implications for both recreational and serious game design.

## Figures and Tables

**Table 1 ijerph-17-08617-t001:** Illustrative quotations from participants regarding avatar use and function (Section 3.1).

Themes/Context	Quote (#)	Participant Characteristics
Primary Theme: Avatar as a Reflection of Experienced Gender		
Subtheme: Creation of avatar as experienced gender	(#1) I’ll tell you about Animal Crossing. It’s like you have your own little town and you’re the Mayor and like picking the male character, it was like this huge deal for me, even though it was just like in private, it was in my bedroom.	Non-binary/male, 20
Subtheme: Creating avatar facilitates gender identity consolidation	(#2) I think creating an avatar is a really good way of solidifying what your personal image of yourself is….I find also avatars are a really great way of testing out who you want to be because a lot of (TGD) people don’t have a solid view of themselves at first….So an avatar as a way of exploration is a really interesting sort of like a thing that I think could be pretty important as well.	Male, 22
Subtheme: Positive emotional impact of using avatar in experienced gender	(#3) Not really having much like experience as a girl, and like, I still was pretty masculine anyway, I wouldn’t necessarily have wanted it, at the same time, in real life but then being able to play as sort of a feminine person in “Life is Strange” it, cause it was mostly a game about their life and like just about being them, it was very, very validating and very nice to be able to go through, sort of going through life as like a cis (cisgender) person or a cis woman, I guess.	Agender, 16

**Table 2 ijerph-17-08617-t002:** Illustrative quotations from participants on their experiences and perspectives regarding avatar design (Section 3.2).

Themes/Context	Quote (#)	Participant Characteristics
Primary Theme: Customisation		
Subtheme: Customisation is key and facilitates gender identity expression and consolidation	(#4) Yeah my Animal Crossing one is one I always come back to and when I think about this sort of thing and kind of the way I felt looking at this character is that this is a character I would like to be and I gave him a name that I was thinking about at the time…..but that was a huge one because that’s me, that’s my insertion into this world and it’s a boy!	Non-binary/male, 20
	(#5) (Referring to a game) so you can hypothetically make your characters trans and there were other characters in it that were confirmed as trans as well so that was a huge thing and it was at a time when I was thinking about these things and I was like “oh my god, I can make my character trans! I can give him a binder!” One of them was a big guy who had a binder so I was like “holy shit, this is something I can choose!”	Non-binary/male, 20
Subtheme: Customization determines playability/purchase	(#6) Like I don’t play The Sims like actual Sims, I just download custom content and just make my characters, so that’s a thing that draws me to a game.	Male, 17
Primary Theme: Avatar Gender Identity		
Subtheme: Customizable or diverse options for pronouns	(#7) It would be great if you had the choice to write your own pronoun so you can customize that, it’s not that hard to program…	Male, 19
Subtheme: Non-binary options	(#8) I know if I found like a game just by myself and then I just opened it and then just found like a non-binary option or something like that, that ‘wow, cool’ like it’d be really surprising like, it’d just make my day, you know.	Male, 18
Primary Theme: Negative Experiences		
Subtheme: Exclusion	(#9) Games like role playing games, when you first start it starts like (choose) if you’re a boy or if you’re a girl, and if you’re like non-binary or gender diverse you have to try and either not play it or pick one and it’s quite annoying to try and do that because you don’t identify as either of them, so I guess that can also cause a bit of, like, definitely if it is a popular game and all your friends are playing it, it’s like should you make the sacrifice to do that or just not do it so I guess that you could also feel left out if it’s a popular game, so that could cause isolation.	Non-binary, 12
Subtheme: Lack of connection	(#10) With school websites, to help kids study, with avatars you can also either choose a boy or a girl and usually with school if you, with being non-binary or gender diverse or something if you haven’t changed your assigned, cause it goes as your assigned gender at, like sex at birth…. the teachers were “oh no you can’t change your avatar” even though the other kids can change their appearance. And it can kind of be kind of dysphoric as well if you have an avatar that’s with the sex or gender that you’re not.	Male, 15

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
