# Peer review of "The Role of the Avatar in Gaming for Trans and Gender Diverse Young People"

_ijerph, 2020, doi:10.3390/ijerph17228617_

Round 1

Reviewer 1 Report

With interest I read the 'avatar' article. I was the more interested as a member of my own family is TGD and experiences lots of comfort in playing games. H/she (being 19 years old) also indicates to profit a lot from the on line contacts h/she has with the relevant game community of TGD people. This last thing is not mentioned in the article, probably as well because it is not researched. I would be curious though if such data were available. These young people feeling happy in the on line world creating their own avatars, but what about their on line (and maybe even off line) communication with other TGD people?

There is a bit of a fulfilling prophecy in the introduction (lines 34-75) where mention is made of relevant research on the topic under concern stressing the positive effects of gaming on -in this case- TGD people. I imagine there might also be studies which are more critical and in general I find this a kind of vacuum in the study (see below as well). Okay, the gaming and avatars have this positive effect on TGD young people but why no attention paid to the real off line world? I mean, these people may feel happy and accepted in the on line world, when they close their computers or devices they are back in the real world again. How does these avatars and this on line community help them then? I am a bit afraid of setbacks and as such the on line world can be an obstacle in the end to the real acceptance of these young people. I saw in line 227-230 that a reference is made to the real world and how the on line avatar world is a kind of testing ground. But I would be more critical here. The on line world is a world of imagination and desired reality and the off line world is everything but that.

I found these articles on line as well, on the negative effects of avatars:

https://www.researchgate.net/publication/228749599_The_Priming_Effects_of_Avatars_in_Virtual_Settings

https://www.sciencedaily.com/releases/2009/11/091110211037.htm

I understand that in this article the use of avatars has a positive effect on TGD young people but the articles I mention here also show that in other contexts avatars can have a negative effect. To be complete I would advise the authors to tackle this issue as well.

I had some more technical remarks on section 2 (materials and methods). When I read lines 87-95 it is not clear to me what experiments the respondents underwent exactly. What happened in these focus groups? What data were transcribed? Was it just a meeting? On what basis were the focus grouped formed? More information please on the methodological procedures would be very much welcome as well as how the transcripts were analyzed. 

Line 113: what or who is AOD again?

What struck me in general as well that no mention is made of the games the young people play. I would be interested to know which they are. Only in Table 2 , #6 a game is mentioned (The Sims). 

Then in lines 218-222 (and 256) the article states that 'this is the first dedicated study....'. I would be a bit more modest because maybe when the article appears there are other studies in the field. But this is a personal thing. 

The were my remarks, I wish the authors all the best and hope they can profit from them.

Author Response

Dear Reviewer 1, 

thank you for your consideration of our manuscript and your valuable comments - I have attached our cover letter addressing your comments accordingly,

very best wishes, Helen Morgan 

Reviewer 1

This last thing is not mentioned in the article, probably as well because it is not researched. I would be curious though if such data were available. These young people feeling happy in the on line world creating their own avatars, but what about their on line (and maybe even off line) communication with other TGD people?

Thank you for your comments. We certainly found participants also derived positive benefits from interacting with other TGD young people online and, as you state, this is a deserving but not well-researched area. Given our focussed interest in the use of the avatar and the length restrictions inherent in producing a paper for publication we elected to concentrate on reporting on this aspect of gaming use solely.

There is a bit of a fulfilling prophecy in the introduction (lines 34-75) where mention is made of relevant research on the topic under concern stressing the positive effects of gaming on -in this case- TGD people. I imagine there might also be studies which are more critical and in general I find this a kind of vacuum in the study (see below as well). Okay, the gaming and avatars have this positive effect on TGD young people but why no attention paid to the real off line world? I mean, these people may feel happy and accepted in the on line world, when they close their computers or devices they are back in the real world again. How does these avatars and this on line community help them then? I am a bit afraid of setbacks and as such the on line world can be an obstacle in the end to the real acceptance of these young people. I saw in line 227-230 that a reference is made to the real world and how the on line avatar world is a kind of testing ground. But I would be more critical here. The on line world is a world of imagination and desired reality and the off line world is everything but that.

Thank you for this valuable comment. While we agree that the online world provides a very different context to the offline world, we note that this study was inductive in nature and consequently our findings were led by the information participants gave us. As reported, they reflected that online use of the avatar to explore, develop and rehearse their experienced gender was positive and helpful in preparing to ‘come out’ in the real or offline world. We did not gather any further data on how it actually impacted in their offline worlds, but we agree this is an interesting line of research for the future and we have now noted this in the discussion section as follows (lines 312-315):

“Further exploration of how the use of the avatar impacts on offline or real world experiences of realizing experienced gender is also worthy of pursuit, particularly in terms of how the experiences of avatar use reported by our participants impacted the potentially challenging process of coming out in the offline world.”

I understand that in this article the use of avatars has a positive effect on TGD young people but the articles I mention here also show that in other contexts avatars can have a negative effect. To be complete I would advise the authors to tackle this issue as well.

With reference to the negative effects of avatar use we thank you for highlighting the study you found. We acknowledge that avatar use can have potentially negative effects however the only studies we could find in this regard pertained to researcher manipulation of the avatar which primed the user to have negative thoughts which is a different phenomenon. This is found in the study you have highlighted which is a different phenomenon in the avatar literature and is beyond the scope of our study.

What struck me in general as well that no mention is made of the games the young people play. I would be interested to know which they are. Only in Table 2 , #6 a game is mentioned (The Sims). 

We elected to exclude the names of the games used by participants given the small, localised, somewhat homogenous sample and the likelihood these choices reflected idiosyncratic interests of participants. If requested we can include an example in the manuscript however we felt that the features of the games played were more important than the games themselves. For your personal interest the games most cited were Animal Crossing, The Sims, Life is Strange, Night in the Woods and Dream Daddy.

I had some more technical remarks on section 2 (materials and methods). When I read lines 87-95 it is not clear to me what experiments the respondents underwent exactly. What happened in these focus groups? What data were transcribed? Was it just a meeting? On what basis were the focus grouped formed? More information please on the methodological procedures would be very much welcome as well as how the transcripts were analyzed. 

This study was qualitative in nature and did not involve experimental manipulation. Participants took part in focus groups using standard focus group processes; specifically young people attended the focus groups and engaged in semi-structured discussions facilitated by a series of pre-determined open ended questions. The verbal data were audio-recorded and transcribed and coded using qualitative coding software (NVivo) to draw out the principal themes. More information regarding the analysis method is provided below. Regarding sampling procedure we draw your attention to lines 87-89 “recruitment occurred via study promotion in local and university LGBT+ services, through social media and an existing longitudinal gender diversity cohort study”. With regard to the nature of data gathering and analysis we have added to the analysis section as follows for further clarity (lines 114-139):

“The aim of the study was to explore key themes in the focus group responses without the imposition of pre-determined premises and develop a model reflecting the structure of participants’ experiences based on the data.  A general inductive approach (GIA), similar to grounded theory and phenomenological approaches, was used to analyze the data and build understanding from observation rather than testing a priori hypotheses [26]. The focus groups were audio-recorded and transcribed verbatim, and the transcripts were then checked against the audio recordings prior to analysis. The transcripts were de-identified by removing participants’ names and any other potentially identifying details. Key themes were identified and coded using NVivo qualitative data analysis software (Version 11; QSR International Pty Ltd., 2015) [27]. After repeated listening of the audio recordings and reading of the transcripts to consider in-text meaning without prior expectation, the following steps, as suggested by Thomas [26] , were taken by HM: 1) specific segments of information were identified; 2) segments were labelled to create categories; 3) similar categories were clustered to reduce overlap and redundancy; 4) a model incorporating the most important categories was created in consultation with co-author AOD.

Quality procedures were used to enhance the trustworthiness of the study findings [28], such as member checks at the end of each focus group, copies of the full transcript made available to participants for comment and a summary of key themes provided to all participants prior to the final round of data analysis for approval and further comment. Other techniques such as a preliminary literature review, audio recording of the groups, and field notes were employed to ensure rigor. The consolidated criteria for reporting qualitative research (COREQ) checklist was used to guide comprehensive reporting of findings and is included as a supplemental file [29]. COREQ is a 32-item checklist used for comprehensive and explicit reporting of qualitative studies that utilise in-depth interviews and focus groups [29].”

Line 113: what or who is AOD again?

We have now clarified that AOD is a co-author by adding “in consultation with co-author AOD (line 129).”

Then in lines 218-222 (and 256) the article states that 'this is the first dedicated study....'. I would be a bit more modest because maybe when the article appears there are other studies in the field. But this is a personal thing. 

In response to this comment, we conducted another literature search on November 1, 2020 and indeed found another study recently published exploring the use of avatars by trans and gender diverse people, however this study used adult subjects. Therefore, we continue to believe this is the first dedicated study exploring this phenomenon in trans and gender diverse young people. Nevertheless, we have updated the manuscript to state that ‘To our knowledge, this is the first dedicated study…(line 278)’.

Reviewer 2 Report

See attached

Author Response

Dear Reviewer 2, 

thank you for your consideration of our manuscript and your valuable comments - I have attached our cover letter addressing your comments accordingly,

very best wishes, Helen Morgan 

In the Introduction section, I think that authors could make a more in deep research about previous research exploring the effects of gender diversity in mental health. There is a lack of studies about this and about adolescence.

Thank you for your comments. While we are constrained by word count and the focus of the study on avatar use we have included the following regarding research on mental health difficulties in gender diverse young people in the introduction (lines 44-47):

“Consequently TGD young people are at particularly high risk of developing mental health difficulties. Nearly three quarters of Australian TGD young people report having a depression or anxiety diagnosis, nearly eighty percent report self-harming, and nearly half have attempted suicide [10].”

In the participants section, did authors check for participants’ IQ or history or previous mental health problems as possible variables mediating results? If not, this should be noted in the limitations section. Also, how was the sampling procedure? In addition, authors may want to consider including standard deviation when describing participant’s age distribution.

Thank you for your comments. This study was qualitative in nature. As such, we were interested in understanding participants’ lived experiences of avatar use in relation to their gender identity, rather than quantitively measuring these experiences or any potentially mediating variables. It was not appropriate to assess these constructs given the study design. 

Regarding sampling procedure we draw your attention to lines 81-83  “recruitment occurred via study promotion in local and university LGBT+ services, through social media and an existing longitudinal gender diversity cohort study” and we have added in “Convenience sample recruitment” to further clarify the sampling procedure. We have added in the sample standard deviation in line 123 “(M = 16.3 years, SD = 2.76)”.

Results: With regards to the results, it is not clear, nor explained, the nature of the analysis. Did authors have previous categories for the analysis. Did authors perform any statistical analysis?

Regarding nature of the analysis we note this is a purely qualitative study and no statistical analyses were performed. We draw your attention to lines 107-111 regarding the rationale for not having pre-determined categories for the analysis “The aim of the study was to explore key themes in the focus group responses without the imposition of pre-determined premises and develop a model reflecting the structure of participants’ experiences based on the data.  A general inductive approach (GIA), similar to grounded theory and phenomenological approaches, was used to analyze the data and build understanding from observation rather than testing a priori hypotheses [26]”.

In keeping with Reviewer 1’s feedback we have added to the analysis section to better explain the nature of the analysis as follows (lines 114-139):

“The aim of the study was to explore key themes in the focus group responses without the imposition of pre-determined premises and develop a model reflecting the structure of participants’ experiences based on the data.  A general inductive approach (GIA), similar to grounded theory and phenomenological approaches, was used to analyze the data and build understanding from observation rather than testing a priori hypotheses [26]. The focus groups were audio-recorded and transcribed verbatim, and the transcripts were then checked against the audio recordings prior to analysis. The transcripts were de-identified by removing participants’ names and any other potentially identifying details. Key themes were identified and coded using NVivo qualitative data analysis software (Version 11; QSR International Pty Ltd., 2015) [27]. After repeated listening of the audio recordings and reading of the transcripts to consider in-text meaning without prior expectation, the following steps, as suggested by Thomas [26] , were taken by HM: 1) specific segments of information were identified; 2) segments were labelled to create categories; 3) similar categories were clustered to reduce overlap and redundancy; 4) a model incorporating the most important categories was created in consultation with co-author AOD.

Quality procedures were used to enhance the trustworthiness of the study findings [28], such as member checks at the end of each focus group, copies of the full transcript made available to participants for comment and a summary of key themes provided to all participants prior to the final round of data analysis for approval and further comment. Other techniques such as a preliminary literature review, audio recording of the groups, and field notes were employed to ensure rigor. The consolidated criteria for reporting qualitative research (COREQ) checklist was used to guide comprehensive reporting of findings and is included as a supplemental file [29]. COREQ is a 32-item checklist used for comprehensive and explicit reporting of qualitative studies that utilise in-depth interviews and focus groups [29].”

Reviewer 3 Report

Thank you for the opportunity to review your paper. It is a well-designed study, and you made it clear that this was a start at looking at the issues surrounding avatars with transgendered and gender diverse youth. Anything I thought of during my reading of your paper ended up being addressed in your discussion. I appreciate the direct quotes you provided, and I believe they were a good representation of the themes that you found in your focus groups.

I don't know if it is necessarily needed to give initials of the research team. I believe you can talk in a more vague manner. For example, in line 101, you could just say "Two of the research team members were known to some participants..." Do you believe this previous knowledge impacted your results at all?

Mostly issues with grammar were found.

Line 35:  delete comma after "birth"

Line 39:  you have brackets around 14, which I believe are not supposed to be there

Line 42:  add comma after "additional challenges"

Line 59:  delete comma after "aspects of self"

Line 91:  change "data was" to "data were"

Line 111:  break into two sentences, adding a period after "categories" and capitalizing "similar"

Line 115:  add comma after [28]

Author Response

Dear Reviewer 3, 

thank you for your consideration of our manuscript and your valuable comments - I have attached our cover letter addressing your comments accordingly,

very best wishes, Helen Morgan 

I don't know if it is necessarily needed to give initials of the research team. I believe you can talk in a more vague manner. For example, in line 101, you could just say "Two of the research team members were known to some participants..." Do you believe this previous knowledge impacted your results at all? 

Thank you for your comments. We have amended line 101(now line 107) by taking out the researchers’ initials and replacing them with “Two researchers”. We don’t believe that these existing relationships impacted upon group interactions.

Mostly issues with grammar were found.

Line 35:  delete comma after "birth"

Line 39:  you have brackets around 14, which I believe are not supposed to be there

Line 42:  add comma after "additional challenges"

Line 59:  delete comma after "aspects of self"

Line 91:  change "data was" to "data were"

Line 111:  break into two sentences, adding a period after "categories" and capitalizing "similar"

Line 115:  add comma after [28]

 We have amended all grammatical issues except for the Line 39 as 14 is an in-text citation, which needs to appear in brackets. We have, however, amended Line 39 to read “half of all mental health disorders emerge during this period [14] to make it easier to read.

Round 2

Reviewer 2 Report

Authors have addressed all my suggestions. I have no further comments.